# Membrane Transporters Involved in the Antimicrobial Activities of Pyrithione in *Escherichia coli*

**DOI:** 10.3390/molecules26195826

**Published:** 2021-09-26

**Authors:** Jesus Enrique Salcedo-Sora, Amy T. R. Robison, Jacqueline Zaengle-Barone, Katherine J. Franz, Douglas B. Kell

**Affiliations:** 1Department of Biochemistry and Systems Biology, Institute of Systems, Molecular and Integrative Biology, University of Liverpool, Crown St., Liverpool L69 7ZB, UK; 2Department of Chemistry, Duke University, 124 Science Drive, Durham, NC 27708, USA; amy.robison@duke.edu (A.T.R.R.); jacqueline.zaengle@duke.edu (J.Z.-B.); 3Novo Nordisk Foundation Centre for Biosustainability, Technical University of Denmark, Building 220, Kemitorvet, 2800 Kgs Lyngby, Denmark

**Keywords:** pyrithione, metal ions, membrane transporters, gram-negative, *E. coli*, Keio collection, copper, zinc, iron

## Abstract

Pyrithione (2-mercaptopyridine-N-oxide) is a metal binding modified pyridine, the antibacterial activity of which was described over 60 years ago. The formulation of zinc-pyrithione is commonly used in the topical treatment of certain dermatological conditions. However, the characterisation of the cellular uptake of pyrithione has not been elucidated, although an unsubstantiated assumption has persisted that pyrithione and/or its metal complexes undergo a passive diffusion through cell membranes. Here, we have profiled specific membrane transporters from an unbiased interrogation of 532 *E. coli* strains of knockouts of genes encoding membrane proteins from the Keio collection. Two membrane transporters, FepC and MetQ, seemed involved in the uptake of pyrithione and its cognate metal complexes with copper, iron, and zinc. Additionally, the phenotypes displayed by CopA and ZntA knockouts suggested that these two metal effluxers drive the extrusion from the bacterial cell of potentially toxic levels of copper, and perhaps zinc, which hyperaccumulate as a function of pyrithione. The involvement of these distinct membrane transporters contributes to the understanding of the mechanisms of action of pyrithione specifically and highlights, more generally, the important role that membrane transporters play in facilitating the uptake of drugs, including metal–drug compounds.

## 1. Introduction

The rise of antimicrobial resistance is a major public health issue [1,2,3,4,5,6,7,8,9,10,11], and understanding its basis is key to overcoming it. While efflux pumps constitute one important source of resistance [12,13,14,15,16,17,18,19,20,21], another is based on enzymes—such as β-lactamases [22,23,24,25,26]—that degrade the β-lactam class of antibiotics. One strategy to combat this mechanism of drug resistance could be to provide a substance that is not itself antibacterial in the absence of any β-lactamase activity but becomes so in its presence. Noting that copper compounds can be significantly antibacterial [27,28,29,30,31,32,33], an embodiment of this strategy was recently proposed that exploited the ability of β-lactamases to activate a prochelator compound (PcephPT, phenylacetamido-cephem-pyrithione) to selectively release pyrithione (2-mercaptopyridine N-oxide) and thereby facilitate its known copper-dependent cytotoxicity [34,35]. In that work, we found that the combination of Cu(II) and pyrithione was indeed highly bacteriotoxic. This toxicity was manifested by the addition of PcephPT only in strains expressing μ-lactamase activity that became tolerant to pyrithione and Cu(II) by the co-addition of the strong but membrane-impermeable Cu(I)-sequestering agent, bathocuproinedisulfonic acid (BCS) [34]. The ability to release pyrithione selectively by μ-lactamase-producing strains provides a compelling approach to pathogen-selective toxicity.

Pyrithione is a bidentate metal-binding agent that forms neutral, lipophilic metal complexes with divalent and trivalent metal ions [36]. These properties, and its ability to facilitate intracellular metal accumulation, categorise pyrithione as an ionophore, to distinguish its biological activity from metal chelators that sequester or deplete cellular metals [37]. The antimicrobial activity of pyrithione, even in its complexed form with Zn(II), is associated with its copper ionophore activity [34,35,38,39,40]. While it is often assumed that metal ionophores diffuse passively across lipid membranes to release intracellular metals, there is no direct evidence to support this mechanism. On the contrary, early reports presumed the need for mediated membrane transport for pyrithione, given its antibacterial efficacy and its low oleyl alcohol:water partition coefficient of 1.07 at 20 °C, pH 7.3 [41].

Since the uptake and efflux of small molecule drugs, including those that are cytotoxic [42,43], requires the intercession of transport proteins [44,45,46,47,48,49,50], one question that arises is how pyrithione (and/or metals with which it interacts) might enter and exit the cytoplasm of target bacterial cells. The existence of the Keio collection of knockout strains of *E. coli* [51,52] (and their over-expression equivalents [53,54]) provides a convenient means to address this question. Thus, in recent work, and because flow cytometry provides a very convenient uptake assay [55], we have used high-throughput flow cytometry to assess the ability of individual transporters to transport fluorescent molecules [56,57,58]. We here use related genetic strategies to assess the contributions of various transporters to the pyrithione uptake and efflux in *E. coli* cells, indicating substantial roles for several. The metal ion effluxers, ZntA and CopA, were found to have significant roles in protecting the bacterial cell against pyrithione toxicity, while the membrane transporters, FepC and MetQ, emerged as potential mediators in pyrithione uptake. The work presented here redirects the assumed passive crossing of pyrithione (similar to the assumption for most small molecules) to a mediated transport across cell membranes.

## 2. Results

In order to screen the membrane protein subset of the *E. coli* Keio knockout collection, we first evaluated the inhibitory concentrations (IC_50_) of pyrithione—in the presence of excess copper—against the reference strain of *E. coli*, BW25113. The IC_50_ value for pyrithione in complex media (Lysogeny Broth (LB)) had a mean of some 6 µM (Table 1), which decreased approximately ten-fold when the concentration of CuCl_2_ increased to 10 µM and higher (Table 1). Although LB is expected to have baseline levels of copper present, the added metal ion clearly sensitised *E. coli* to the pharmacological actions of pyrithione, consistent with prior studies [34].

### 2.1. Screening Membrane Protein Genes of the E. coli Keio Collection: Identification of Membrane Proteins Potentially Involved in the Pharmacological Actions of Pyrithione

In order to identify membrane proteins that affect pyrithione activity, growth rates in the exponential phase were recorded for the 532 Keio knockout strains lacking individual membrane proteins (Keio subset, Appendix A) in the presence of a single, sublethal dose of 2 µM pyrithione. The effect of pyrithione was quantitated as the ratio of the growth rate for each strain in the presence of this metal ionophore over the growth rate in its absence. As expected, the reference strain of *E. coli*, BW25113 (denoted here as “wild type” (WT)), showed a growth rate ratio with a mean close to 1.0, indicating that no differences in growth are detectable in the exposure versus the control cultures under these conditions (Table 2). The range of ratios for the Keio subset had an asymmetrical, right-skewed distribution with strains at the upper end growing at twice the rates in the presence of pyrithione (e.g., *fepC* and *metQ*, Figure 1). In the lower end of the distribution were strains that grew at half the rate with pyrithione (e.g., *copA*, Figure 1). The largest effect was observed in the strain lacking the *zntA* gene, with growth rates reduced approximately ten-fold when in the presence of sublethal concentrations of pyrithione (Figure 1).

### 2.2. Growth Kinetics of Knockout and Overexpression Strains for metQ, fepC, copA, and zntA Genes at Variable Concentrations of Pyrithione Expanded on the Observations from the Initial Knockout Library Screening

To gain more insight into the roles of the four major hits from the Keio screen with a single dose of pyrithione (2 µM), the growth of the *ΔcopA*, *ΔzntA*, *Δ**metQ*, and *ΔfepC* strains was next measured at the following incremental concentrations of pyrithione: 0, 3, 6, 12, and 25 µM. In addition to the knock-out strains, the over-expression (Aska) strains of these four genes were also challenged with increasing pyrithione and monitored for growth over time.

As observed in the initial screening experiment, the knockouts of *copA* and *zntA* sensitised the cells to growth inhibition by pyrithione (Figure 2c and Figure 3c). The lack of *copA* had a reduced growth (half of that of the reference strain) at low concentrations of pyrithione with total inhibition at the higher concentrations tested (12 and 25 µM) (Figure 2c). The absence of *zntA* resulted in even more severe growth inhibition, with cell growth halted from the lowest concentration tested (3 µM) (Figure 3c).

Overexpressing *copA* (Aska_copA) was visibly detrimental, with the duration of its lag phase being double that of the reference strain (Figure 2a). The impaired growth of Aska_copA was observed throughout 24 h (Figure 2a). Under the presence of pyrithione, Aska_copA had growth trends proportionally similar to that of the control for the initial concentrations of pyrithione. From 12 µM pyrithione, Aska_copA did not grow. The overexpression of *zntA* (Aska_zntA) was also detrimental for growth compared to the parental strain (Figure 3a), although not as severely as Aska_copA. In the presence of pyrithione, the Aska_zntA growth was slightly impaired compared to the parental strain, but noticeably more resistant to pyrithione compared to the *zntA* knockout (Figure 3c).

The growth of the knockout strains for the ABC transporters MetQ and FepC in the absence of pyrithione was comparable to the parental strain in both the lag phase and the growth rate (Figure 4c and Figure 5c). The growth at the sublethal concentration of 3 µM pyrithione showed *Δ**metQ* and *ΔfepC* growing at equivalent levels with the absence of pyrithione, but not the higher levels that were observed for these strains in the large set screening (Table 2). We attribute that dissimilar behaviour to different conditions of nutrient circulation and partial oxygen pressures present in the wells of 384-well plates versus those conditions of much larger containers used for individual cultures when dealing with a discrete number of strains (i.e., only the reference strain plus *Δ**metQ, ΔfepC, ΔcopA* and *ΔzntA*). Nonetheless, the growth of *ΔfepC* at higher (≥6 µM) concentrations of pyrithione did supersede that of the parental strain (Figure 5c).

The expression of multiple copies of the gene encoding for MetQ (Aska_metQ) impaired *E. coli* growth, with approximately two-fold longer lag phases in complex media in comparison to the reference strain BW25113, which contains only the chromosomal copy of the same gene (Figure 4a). Under incremental concentrations of pyrithione, the overexpression of *metQ* consistently displayed longer lag times and lower growth rates in the exponential phase compared to the equivalent in the reference strain. The cell growth was then assessed in media with pyrithione and MetQ’s natural substrate, methionine, in order to discriminate MetQ-mediated pyrithione effects from confounding effects of the impaired growth fitness brought by the overexpression of MetQ. In complex media that had excess methionine, MetQ overexpression exacerbated sensitivity to pyrithione (i.e., three-fold longer lag phase with 6 µM pyrithione compared to two-fold in the control strain, Figure 4). The growth rates in exponential phases were again consistently lower in the strain expressing high copies of *metQ* than in the reference strain.

The overexpression of *fepC* (Aska_fepC) also caused an increased lag phase by approximately two-fold in comparison to the reference strain (Figure 5a). This effect was consistently observed through incremental concentrations of pyrithione until the growth for this strain was halted at 12 µM pyrithione. The increased susceptibility of the Aska_fepC and Aska_metQ strains supports the notion that these transporters could be a pathway by which pyrithione accesses the cytosol.

### 2.3. Effects of the Gene Knockouts on the Minimum Inhibitory Concentrations of Pyrithione in the Presence and Absence of Relevant Metals

Given the notable metal involvement for three of the four genes identified in our knock-out hits, we assessed the minimal inhibitory concentration (MIC) of pyrithione in the absence and presence of supplemental Cu(II), Zn(II), and Fe(III), as well as the membrane impermeable Cu(I)-sequestering agent, BCS. The MICs of pyrithione alone were lowest for *ΔzntA* (2.2 µM) and *ΔcopA* (8.8 µM), compared to the others at 17.5 µM (Table 3 and Figure 6a). Adding Cu did not shift the MIC for WT, *ΔmetQ*, or *ΔfepC,* but did lower it for *ΔcopA* and slightly increased it for *ΔzntA*. Adding Zn had the opposite effect, by decreasing the MIC of *ΔzntA* but increasing it for *ΔcopA.* The other strains were mostly unaffected by Zn, although *ΔmetQ* did result in an MIC lower by one dilution factor (Table 3). Adding Fe, on the other hand, had no effect on the MIC of any of the strains.

The presence of BCS rescued the growth of all the strains tested, as indicated by the increase in the MIC at or above 140 µM pyrithione (Table 3 and Figure 6b). Interestingly, BCS had a diminished effect on the *ΔzntA* strain compared to the others. Although it caused the MIC value to increase from 2 to 140 µM, the growth of the *ΔzntA* strain was significantly lower than the other strains starting at all concentrations of pyrithione (Figure 6b). As BCS is selective for Cu(I) and is not expected to directly affect Zn speciation in these experiments, the abrogation of pyrithione’s activity by BCS points to a Cu-dependent mechanism of toxicity. The weaker effect of BCS on the *ΔzntA* strain, however, suggests a contributing role for Zn as well. To test this idea, we added supplemental Zn in combination with BCS and pyrithione. This combination had a profound effect only on the *ΔzntA* strain, with the MIC value that decreased from the BCS-induced 140 to 17.5 µM PT upon the addition of Zn (Table 3 and Figure 6c).

As the MIC values alone are not an indication of cell killing, the bactericidal effects of pyrithione were measured for all five strains in the presence of either Cu, Fe, or Zn, as indicated in Figure 7. Pyrithione alone is not bactericidal, although the absence of the ZntA transporter was more sensitive than any of the other strains (Figure 7). The presence of excess copper made pyrithione cytocidal to all five strains. In agreement with the growth data (Figure 4), the absence of MetQ showed some level of protection in some of the samples (Figure 7). When the cells were treated with zinc or iron in addition to pyrithione, *ΔzntA* had the most samples that showed partial cidality. The presence of BCS, though, seemed to decrease the cytocidal effects of pyrithione for *ΔzntA*. Taken together, these data reinforce the notion that Cu is the strongest mediator of pyrithione toxicity, with Zn perhaps playing a supporting role.

### 2.4. Levels of Cell-Associated Metals in the Absence of Membrane Transporters CopA, ZntA, FepC, and MetQ

In order to assess the relative effects of pyrithione on the total cellular metal content of our strains of interest under conditions where the cells are viable, we selected treatment conditions below the MIC values for each strain: *ΔcopA* and *ΔzntA* were treated for 15 min with 0.5 µM pyrithione in the absence or presence of supplemental 10 µM Cu(II), Zn(II), or Fe(III) added to the broth, while *ΔfepC* and *ΔmetQ* were treated similarly, but with 4 µM pyrithione. The copper levels associated with the cells (membranous compartment and/or intracellular) were generally higher in the *ΔcopA* and *ΔzntA* knockouts compared to wild type across all the treatments, with the most notable increase seen in the combination treatment of pyrithione with added Cu (Figure 8a). The total zinc levels were also generally higher on average for *ΔcopA* and *ΔzntA* compared to WT, although the variability in the data makes these changes hard to distinguish (Figure 8b). Notable, however, is the observation that the combination of PT and Zn does *not* lead to Zn hyperaccumulation in the transporter knockouts compared to WT (Figure 8b), in direct contrast to the trend observed for Cu. These data show that the increased sensitivity of the *ΔzntA* strain is not related to an accumulation of Zn, but rather suggest that *zntA* contributes to Cu clearance under pyrithione-induced stress. This conclusion is consistent with the sensitivity of the MIC of this strain to BCS (Table 3). Fe levels in *ΔcopA* and *ΔzntA* were not noticeably affected by pyrithione under the tested conditions (Figure 8c), consistent with expectations based on known substrates for these transporters, which do not include Fe.

The comparison of the *ΔfepC* and *ΔmetQ* strains with wild type reveals interesting metal-dependent responses. The total metal levels increased noticeably for *ΔmetQ* compared to the WT cells subjected to the same combination treatment of pyrithione and Cu, Zn, or Fe (Figure 8d–f). In contrast, the levels of Cu and Zn were unchanged in *ΔfepC* compared to WT when these metals were co-administered with pyrithione (Figure 6d,e), whereas the Fe levels decreased compared to WT for that condition (Figure 6f). The consistent increase in Cu, Zn, and Fe levels when each metal was co-administered with pyrithione suggests something unique about the pyrithione-enabled uptake of these metals when methionine transport is disabled by *ΔmetQ*. Although the data do not provide direct evidence, it is interesting to speculate a potential route for pyrithione-metal complexes via the enterobactin uptake pathway. Notably, when that pathway is prevented by *ΔfepC*, the total Cu, Zn, and Fe levels are not elevated, compared to untreated or metal-only WT controls. These data, therefore, suggest a potential model in which pyrithione is preferentially taken up as an uncomplexed monomer by the methionine transport machinery, while metal complexes preferentially co-opt the siderophore pathway, which would thereby lead to increased cellular metal loads.

## 3. Discussion

Using genetically modified strains of the Gram-negative *E. coli,* we have presented here a discrete number of membrane transporters likely involved in the facilitated transit in and out of the cell of pyrithione and its associated metal cations, most relevantly Cu and Zn. The properties of pyrithione as a metal ionophore are necessary for its cytotoxicity, as originally reported [59], at concentrations similar to those reported here for *E. coli*, particularly in the presence of added Cu (Table 3 and Figure 7, WT strain). At millimolar concentrations, much higher than used here, pyrithione has been shown to inhibit certain membrane transport systems, depolarise the cell membrane in microorganisms, as well as reduce cellular protein synthesis and ATP levels [39,60]. The finding that the *ΔatpA* strain was hypersensitive to low concentrations of pyrithione (Table 2) is consistent with these prior observations. Furthermore, several knockout strains lacking sugar transporters were identified here as having growth impaired by pyrithione (*fucP*, *malF*, *xylE*, Table 2). Combined, the susceptibility of these knockout strains points to the metabolic and respiratory strain experienced by cells as a consequence of pyrithione exposure. Other hits (Table 2 and Appendix A) are represented by “y-genes” (genes of unknown function) that still represent some 35% of the *E. coli* gene complement. The behaviour of the top four knockouts (*metQ*, *fepC*, *copA*, and *zntA*) merited further investigation.

The increased cytotoxicity of pyrithione with added Cu suggests that Cu either exacerbates pyrithione’s direct impairment of these respiratory processes, or Cu could be the direct cause. Evidence for Cu’s toxic effects is abundant, and its direct impact on the proteins required for metabolism, respiration, and protein synthesis is steadily emerging, as revealed, for example, by a study in *Staphylococcus aureus* that found that Cu stress targets proteins involved in central carbon metabolism [61]. Additionally, a recent proteomics study from one of us used protein folding stability measurements to identify protein hits of pyrithione in combination with Cu [33]. In that study, we found a number of protein hits involved in glycolysis, the citric acid cycle, and protein biosynthesis that were distinctly driven by the Cu delivered to the cell by pyrithione [33]. In the context of the current results identifying the *ΔcopA* knockout strain as highly sensitive to pyrithione, a picture emerges that pyrithione’s mechanism of action is directly linked to its facilitation of Cu gaining unregulated access to the cytoplasm of *E. coli*, consistent with its activity in yeast [62].

Copper ions are required in low concentrations for multiple metalloenzymes (e.g., copper oxidases, amine oxidases, cytochrome oxidases, Cu, Zn-SOD), which in *E. coli* are found exclusively in the cell envelope. Bacteria sustain these low levels of the otherwise toxic free cellular copper with vectorial transmembrane transport and compartmental trafficking via cytoplasmic and periplasmic chaperones, avoiding the presence of free, uncomplexed copper [63,64]. Although the entry route for copper is unclear in bacteria, its efflux relies on metal exporters such as CopA, a P-type ATPase that sits on the inner membrane and actively clears the cytosol of Cu(I) by pumping it into the periplasm, where it can subsequently be oxidised to the less damaging Cu(II) by CueO, incorporated into cuproenzymes, or effluxed from the cell by the Cus system [65,66,67,68]. CopA is a Cu(I) exporter specifically induced by copper or silver [67,68]. CopA interacts with soluble chaperones on either side of the cytoplasmic membrane as part of the ion permeation path that also includes the outer membrane CusCBA system for the efflux of copper to the extracellular milieu [69,70,71]. 

The absence of CopA decreases the cytoplasmic efflux efficiency, thereby leaving the cytoplasm susceptible to the toxicity of Cu(I) [70]. High concentrations of Cu can induce unregulated membrane permeability and impair proton-coupled membrane transport, which increases the lag growth phase duration and reduces the rate of exponential growth [72], in line with our observations on the effect of pyrithione on the *ΔcopA* knockout strain (Figure 1 and Figure 2c, Table 3). The absence of *copA* increased the concentrations of cell-associated Cu (Figure 8a) and the cytotoxicity of pyrithione diminished when Cu was sequestered by the membrane-impermeable Cu(I) chelator BCS (Figure 6 and Figure 7). Overexpressing CopA did not overcome the toxicity of pyrithione as we might have expected given its role (Figure 2a). However, the overexpression of membrane transporters is known to be mostly deleterious [73,74]. Pyrithione itself has been shown to induce the overexpression of CopA and CueO [33]. It could be that excess CopA, as provided genetically in the Aska-CopA strain, might have impaired the homeostatic machinery to such a degree that the exposure to pyrithione only worsened downstream cytotoxicity. 

While the data collected in this study consistently point to a primary role for Cu toxicity in the mechanisms of pyrithione exposure, the fact that the *ΔzntA* knockout strain provided the strongest growth-impaired phenotype raises the question of Zn’s contribution to pyrithione activity (Figure 1 and Figure 3c, Table 2). The most conclusive evidence for a direct role for Zn was observed by the re-sensitisation of *ΔzntA* to pyrithione when Zn was added to the BCS-treated cells (Table 2, Figure 6). BCS treatment creates a functional Cu deficiency that abrogated pyrithione activity in the other strains tested, with the MICs increasing to 140 µM or higher. In contrast, BCS did not recover *ΔzntA* growth to the same extent (Figure 6b), and this strain was re-sensitised when Zn was added in combination with pyrithione and BCS (Figure 6c). ZntA is a Zn(II) and cadmium(II) P-type ATPase [75,76], with Cd(II) being its most effective inducer [77,78,79]. Its transport activity for Cd(II) or other non-native but known heavy metal substrates such as Pb(II) or Hg(II) is not expected to be operative under the conditions of these experiments, although it is possible that ZntA is contributing directly to Cu clearance. It is also possible that Cu stress could cause the release of internal Zn stores that become problematic when ZntA is missing. Interestingly, ZntA was one of the strongest over-expression and protein-stability hits from our previous proteomics study [33]. The analysis of these data suggested that pyrithione, not Cu, drove the increase in protein stability. An increase in protein stability is induced by ligand binding, which, in this case, could be driven by a metal binding to ZntA, or alternatively, an adduct with pyrithione itself, either as a ternary complex of transporter-metal-pyrithione or as a covalent modification such as disulfide bond formation between pyrithione and one of the many cysteines of ZntA (and CopA).

The identification of CopA and ZntA as critical components of the resistance mechanism against pyrithione-induced toxicity is consistent with the functional annotation of pyrithione as an ionophore that facilitates metal hyperaccumulation in cells. These transporters, however, do not address how pyrithione and its associated metals gain access to the cytoplasm to induce toxicity. This study identified MetQ and FepC as likely candidates, although it is important to note that these are not expected to be exclusive routes for pyrithione uptake, as evidenced by the number of other gene knockouts that showed increased growth rate ratios with pyrithione (Table 1). Nevertheless, it is interesting to speculate whether MetQ and FepC facilitate the uptake of pyrithione as a monomer or as a metal complex.

The lipoprotein MetQ is the substrate binding protein for MetNI, the ATP Binding Cassette (ABC) membrane transport system responsible for the high affinity import of methionine [80,81]. MetQ forms a stable complex with the rest of this transport system via an N-terminal lipidic modification [82]. The observation that the *metQ* knock-out strain grew comparatively better in the presence of pyrithione suggests that the cells gained some growth advantage in the presence of this sulphur-containing molecule when the cells were deprived of their regular methionine-uptake system. The absence of *fepC* also provided a noticeable advantage under these conditions. FepC is the cytoplasmic membrane transporter that, together with a periplasmic protein (FepB) and an outer membrane transporter (FepA), are the ABC membrane transport complex that facilitates siderophore-mediated iron uptake using the siderophore enterobactin [83]. The identification of a transporter involved in taking up a metal complex raises the question of whether the metal coordination complexes of pyrithione might co-opt this system to gain entry to the bacterial cytoplasm. 

A number of other hits that decreased the growth rate ratio are involved in sugar transport (*fucP*, *malF*, *xylE*) and energy storage (*atpA*) (Table 2). The susceptibility of these strains signals that pyrithione is causing strain on metabolic processes. Other hits (Table 2 and Appendix A) are represented by “y-genes” (genes of unknown function) that still represent some 35% of the *E. coli* gene complement. The behaviour of the top four knockouts (*metQ*, *fepC*, *copA* and *zntA*) merited further investigation. 

The MetQ-mediated influx of methionine seemed to have increased the penetration of pyrithione (Figure 4d), which points to a co-transport mechanism for these two molecules. It is also possible that Cu(I) could highjack this uptake pathway by binding to pyrithione and/or the thioether group of methionine [64], a natural substrate of MetQ. Both scenarios, methionine and pyrithione co-transport by MetQ, and the affinity of MetQ for substrates that can form metal complexes, could explain the hyper-sensitivity to pyrithione in the presence of excess methionine in cells overexpressing *metQ* (Figure 4d). Other transport systems might be involved in this methionine-dependent sensitivity to pyrithione since a similar but lesser effect was also seen in the reference strain and the knockout strain for *metQ* (Figure 2e,f). The likely candidates would be expected to be in the strains that survived the exposure to sublethal pyrithione similarly or better than the reference strain (Table 2).

We also observed that the strain lacking *metQ* had higher cell-associated Cu and Zn in the presence of pyrithione vs. the metal alone compared to the other strains included in this study (Figure 8d–f). Cell-associated Cu can exist in two oxidation states, Cu(I) and Cu(II), which are readily complexed by biological ligands. Cupric ion, Cu(II), can exist in the oxidising environment of the periplasm, while the more reactive cuprous ion, Cu(I), is the more likely form in the reducing environment of the cytoplasm [64]. The ICP-MS data collected here for total cell-associated metal levels do not distinguish the compartmental location or the oxidation state. The observation of augmented concentrations of cell-associated metals in *metQ* knockout cells without seeing an increased susceptibility to pyrithione suggests that this extra metal load might be accumulating in the periplasm. The chelation of metals outside the cytoplasm by the accumulated pyrithione (e.g., in *metQ* knockout cells) might not generate cytotoxicity. An alternative scenario is that MetQ transports pyrithione alone, with pyrithione–metal complexes entering via an alternate route, such as FepC. In this scenario, the absence of MetQ could increase the flux through those alternates, resulting in higher total metal levels.

As with *metQ*, the absence of *fepC* also provided a noticeable advantage under these conditions. FepC is the cytoplasmic membrane transporter that together with a periplasmic protein (FepB) and an outer membrane transporter (FepA) are the ABC membrane transport complex that facilitates siderophore-mediated iron uptake using the siderophore enterobactin [83]. The identification of a transporter involved in taking up a metal complex raises the question of whether metal coordination complexes of pyrithione might co-opt this system to gain entry to the bacterial cytoplasm. The overexpression of *fepC* sensitised *E. coli* to the growth inhibition caused by pyrithione, while its absence protected the cells to this effect, Figure 5a,c, respectively. The *fepC* knockout strain only presented cell-associated metal changes for iron. In the absence of fepC, the levels of iron were lower than the reference strain, and more visibly, they were also lower than the *metQ* knockout strain (Figure 8f). Pyrithione seemed to have decreased further the level of iron in the fepC knockout. Altogether, this implies FepC transports pyrithione:iron complexes that, rather straightforwardly, follows its natural role as a transporter of iron complexes [83].

## 4. Conclusions

These findings further our understanding of the cellular activity of pyrithione. Our data are consistent with a model of pyrithione complexing Cu(II), then transporting and releasing the more reactive Cu(I) intracellularly. Our data suggest that this ionophore activity is mediated via membrane transport as opposed to passive diffusion. This mechanism was envisaged in the early characterisation of pyrithione’s chemistry and biological toxicity in bacteria that indicated that the compound forms chelate complexes and exerts its action intracellularly following dissociation. A likely mediated membrane transport—reported here—for crossing was logically anticipated as “…a third factor playing a part” [41]. Altogether, we envisage the flow of a metallodrug such as pyrithione that can chelate cations on either side of the plasma membrane and is transported in or out of the cell by host proteins with or without its metal cargo.

## 5. Materials and Methods

### 5.1. Strains and Culture

The *E. coli* Keio collection of gene knockout strains was provided by the National Institute of Genetics, Mishima, Shizuoka, Japan [51,52]. A subset of 532 strains from the Keio Collection whose cognate knockout genes are annotated as encoding for membrane proteins were selected for this study (Appendix A). Individual knockout strains highlighted in this work are as follows: ***copA***
*(F-, Δ(araD-araB)567, ΔlacZ4787(::rrnB-3), λ-, ΔcopA767::kan, rph-1, Δ(rhaD-rhaB)568, hsdR514); **fepC** (F-, Δ(araD-araB)567, ΔlacZ4787(::rrnB-3), ΔfepC726::kan, λ-, rph-1, Δ(rhaD-rhaB)568, hsdR514); **metQ** (F-, Δ(araD-araB)567, ΔlacZ4787(::rrnB-3), λ-, ΔmetQ722::kan, rph-1, Δ(rhaD-rhaB)568, hsdR514); **zntA** (F-, Δ(araD-araB)567, ΔlacZ4787(::rrnB-3), λ-, rph-1, ΔzntA724::kan, Δ(rhaD-rhaB)568, hsdR514)*. PCR was used to verify that we were given the correct strains. We also used strain overexpressing *copA*, *fepC*, and *metQ*, and *zntA* from the ASKA collection: *E. coli* K-12, strain AG1 [*recA1 endA1 gyrA96 thi-1 hsdR17 (r K− m K+) supE44 relA1*] carrying recombinant constructs in the IPTG-inducible and multicopy plasmid pCA24N (*CmR, lacIq*). The induction with IPTG was optimised to 250 µM for 3 h (37 °C, shaking at 200 rpm) before the growth assays. The ASKA collection [53] was also provided by the National Institute of Genetics, Mishima, Shizuoka, Japan.

Pyrithione (2-mercaptopyridine N-oxide) was purchased from Sigma (Cat 188549) and stock solutions were prepared in dimethyl sulfoxide (DMSO). Bacterial cultures were routinely carried out in lysogeny broth (Merck LB 110285) [58]. Growth inhibitory concentrations (IC_50_) for pyrithione in the *E. coli* Keio reference strain BW25113 were calculated from microtitration assays. Overnight cultures in LB were diluted 1 in 1000 in fresh LB. Fifty µL of two-fold dilutions of pyrithione were transferred to 96-well plates to which 50 µL of fresh bacterial culture were added. Endpoint reads of the media turbidity at 600 nm (OD_600_) were taken after 24 h at 37 °C.

The pyrithione exposure growth assays for the Keio collection subset of 532 strains (Appendix A) were carried out by replicating their glycerol stocks into 384-well plates with 50 µL of LB containing 30 µg/mL of kanamycin. These sealed plates were incubated overnight at 37 °C. The overnight cultures were diluted 1 in 1000 in 50 µL of fresh LB without kanamycin in polystyrene 384-well plates with a transparent bottom (Sigma M6936-40EA). These plates were sandwiched with plastic covers (CR1384, Enzyscreen) and incubated for 24 h at 37 °C with 225 rpm shaking in the Growth Profiler. Plates with pyrithione contained this compound at 2 µM final concentration.

### 5.2. Growth Profiling

The Growth Profiler 960 (Enzyscreen, NL; http://www.enzyscreen.com/growth_profiler.htm, accessed on 16 June 2020) records the pixel density (G values) of photographs per well. This instrument estimates culture density using the changes in these camera-based measurements, in parallel, of up to ten 96-well plates by extrapolating G values into optical densities (OD_600_) using strain and media-dependent standard curves [57]. Although 384-well plates are not the usual format used for the Growth Profiler, we observed that the G values from image analyses provided sigmoidal curves over time that fitted into non-linear parametric regressions used here to calculate exponential growth rates. The growth of smaller number of strains was also measured by media turbidity (OD_600_) in the plate reader BMG Clario plus (BMG).

The inhibitory concentrations of pyrithione that kills half of the bacteria population (IC_50_) were calculated with the four-parameter logistic model as implemented in the R [84] and the package *drc* [85], following the guidelines for relative calculations according to the spread of the data [86]. The growth rates in the exponential phase of batch cultures were extracted from the curve fitting of cell growth with a parametric non-linear method (i.e., method of least squares). The lag times were calculated using a linear fitting of the log-transformed data [87,88]. Both approaches were carried out as implemented in the R package *growthrates* [89].

For the microdilution assays, the Keio knockout strains were grown for approximately 20 h on LB agar containing 50 µg/mL kanamycin. The wild type was grown for approximately 20 h on LB agar. One colony of the respective knockout strain was used to inoculate 2 mL of LB containing 50 µg/mL kanamycin. The wild type was inoculated into 2 mL of just LB. These were then incubated for 16–18 h at 37 °C and 200 rpm. The overnight cultures were then diluted 1:500 in fresh LB and used as the working solution. Pyrithione was serially diluted 2-fold in LB to reach concentrations of 0–140 µM and plated in a clear-bottomed 96-well plate. The cell working solution was then treated with 10 µM CuCl_2_, ZnCl_2_, ferric ammonium citrate, and/or the copper(I) chelator bathocuproinedisulfonic acid (BCS) and added to the plates for a final dilution of 1:1000 and 100 µL per well. Wells containing media only, cell working solution only, and treated cell working solution only, were included to verify sterility and the effect of metals alone. Plates were incubated at 37 °C and 200 rpm for 20 h. The plates were sealed with AeraSeal film (EXCEL Scientific) to minimise evaporation during incubation. Bacterial growth was determined by measuring the OD_600_ using the PerkinElmer Victor3 V multilabel plate reader at 0 and 20 h. The 0-h time point was subtracted from the 20-hour time point to remove the background. The minimum inhibitory concentration (MIC) was defined as the dilution at which no more growth occurred (20 h OD_600_ < 0.010). Three biological replicates were performed, each with 3 technical replicates. MICs were measured after bacteria were incubated at 37 °C and 200 rpm in LB for 20 h. Growth was assessed using OD_600_ measurements. The following PT concentrations were tested: 140, 70, 35, 17.5, 8.8, 4.4, 2.2, 1.1, and 0.5 µM. MICs within one dilution factor were not considered statistically significant. A minimum of 2 biological replicates were performed. Each biological replicate had a minimum of three technical replicates. 

At the 20-h time point of the microdilutions above, 4 µL of each of the MIC-treated cultures were spotted on fresh LB agar and allowed to dry. These were then incubated at 37 °C and 200 rpm for 24 h. The number of colonies grown were then counted. 

### 5.3. Measurement of Cell Associated Copper

Intracellular metal concentrations were determined using inductively coupled plasma mass spectrometry (ICP-MS). The Keio knockout strains were grown for approximately 20 h on LB agar containing 50 µg/mL kanamycin. The wild type was grown for approximately 20 h on LB agar. One colony of the respective knockout strain was used to inoculate 2 mL of LB containing 50 µg/mL kanamycin. The wild type was inoculated into just 2 mL of LB. The overnight cultures were then diluted 1:1000 and grown to an OD~0.1. The resulting culture was then aliquoted into metal-free falcon tubes. The aliquoted cultures were treated with 0.5 or 4 µM PT and/or 10 µM CuCl_2_, ZnCl_2_, ferric ammonium citrate for 15 min at 37 °C, 200 rpm. Untreated samples were included for comparison. The cultures were centrifuged at 1000× *g* and 4 °C for 5 min. The cells were then washed twice with 1 mL of metal-free water and once with 1 mM EDTA. Cells were dried overnight at 80–90 °C. Cells were digested in 100 µL of concentrated trace metal-grade nitric acid for 1–2 h. Then, 900 µL of 1% trace metal-grade nitric acid was added, and the samples were stored at room temperature until ICP-MS analysis. ICP-MS analysis was performed in the OHSU Elemental Analysis Core using an Agilent 7700x equipped with an ASX 500 autosampler. The system was operated at a radio frequency power of 1550 W, an argon plasma gas flow rate of 15 L/min, and Ar carrier gas flow rate of 0.9 L/min. Elements were measured in kinetic energy discrimination (KED) mode using He gas (4.3 mL L/min). Data were quantified using weighed, serial dilutions of a multi-element standard (CEM 2, (VHG labs, VHG-SM70B-100) Mn, Fe, Cu, Zn) and a single element standard for P (VHG labs, PPN-500). For each sample, data were acquired in triplicates and averaged. A coefficient of variance (CoV) was determined from frequent measurements of a sample containing ~10 ppb. An internal standard (Sc, Ge, Bi) continuously introduced with the sample was used to correct for detector fluctuations and to monitor plasma stability. Accuracy of the calibration curve was assessed by measuring NIST reference material (water, SRM 1643f) twice during the measurement and found to be within ±10% for all determined elements. Data were normalised to the concentration of phosphorus for each sample.

## Figures and Tables

**Figure 1 molecules-26-05826-f001:**
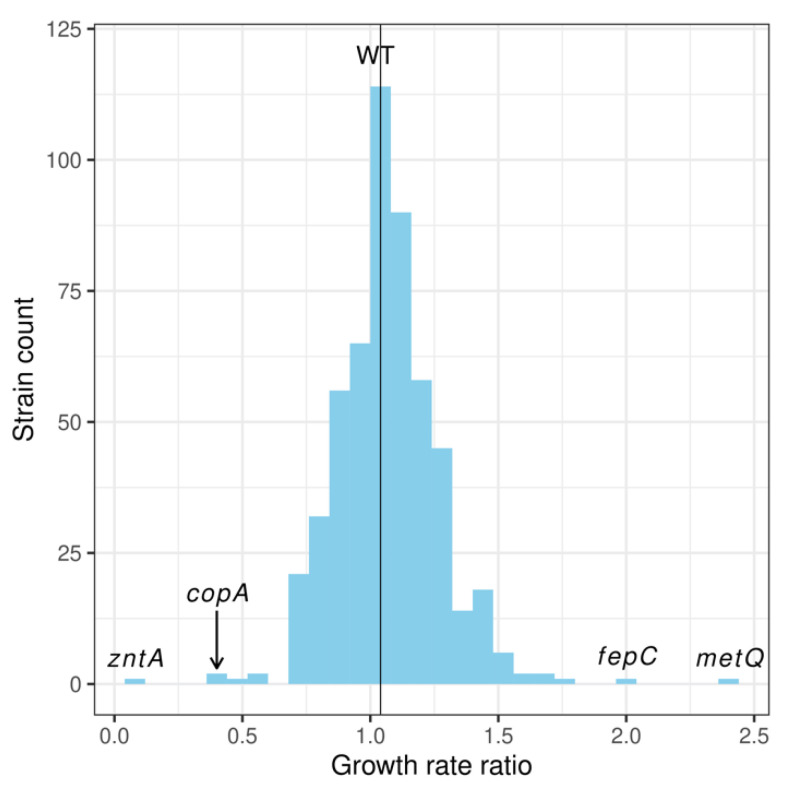
**Distribution of bacterial growth in the presence of 2 µM pyrithione**. The Keio collection subset (532 knockout strains for gene encoding membrane proteins) was exposed to pyrithione and their growth compared against the unexposed samples and these ratios are represented in the abscissa. The reference strain (WT) with a mean ratio of 1.04 occupies the bin that contains the highest count of strains. The following four boundary knockout strains selected here are further denoted by the location of their growth rate ratios of exposed over unexposed: *zntA*, *copA*,* fepC*, and *metQ*.

**Figure 2 molecules-26-05826-f002:**
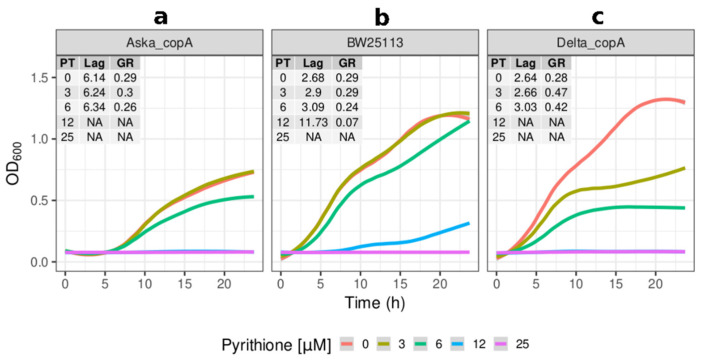
**Effects of *copA* on the sensitivity of *E. coli* to pyrithione.** Growth (OD_600_) at incremental concentrations of pyrithione in complex media (LB) of the (**a**) Aska clone carrying episomal copies of *copA* (Aska_copA), (**b**) the reference strain BW25113, and (**c**) the *copA* knockout strain (Delta_copA). Insets: PT: concentration of pyrithione in micromolar (µM), Lag: length of lag phase in hours, GR: growth rate fitted from optical densities of the cultures in exponential phases. The colour coded lines represent the fitted data with the parametric non-linear method of least squares. NA (non-applicable): data below detection limit.

**Figure 3 molecules-26-05826-f003:**
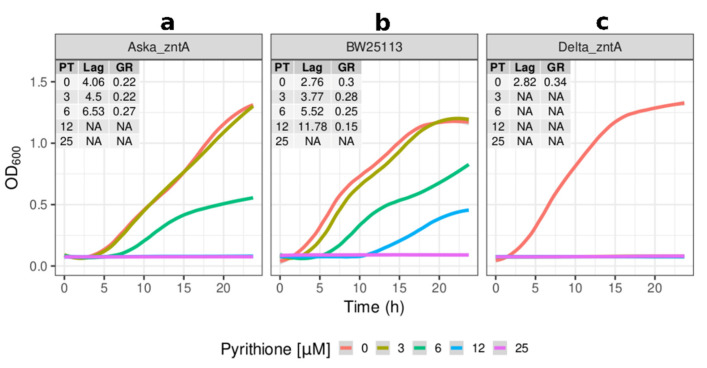
**Effects of *zntA* in the sensitivity of *E. coli* to pyrithione.** Growth (OD_600_) at incremental concentrations of pyrithione in complex media (LB) of the (**a**) Aska clone carrying episomal copies of *zntA* (Aska_zntA), (**b**) the reference strain BW25113, and (**c**) the *zntA* knockout strain (Delta_zntA). Insets: PT: concentration of pyrithione in micromolar (µM), Lag: length of lag phase in hours, GR: growth rate fitted from optical densities of the cultures in exponential phases. The colour coded lines represent the fitted data with the parametric non-linear method of least squares. NA (non-applicable): data below detection limit.

**Figure 4 molecules-26-05826-f004:**
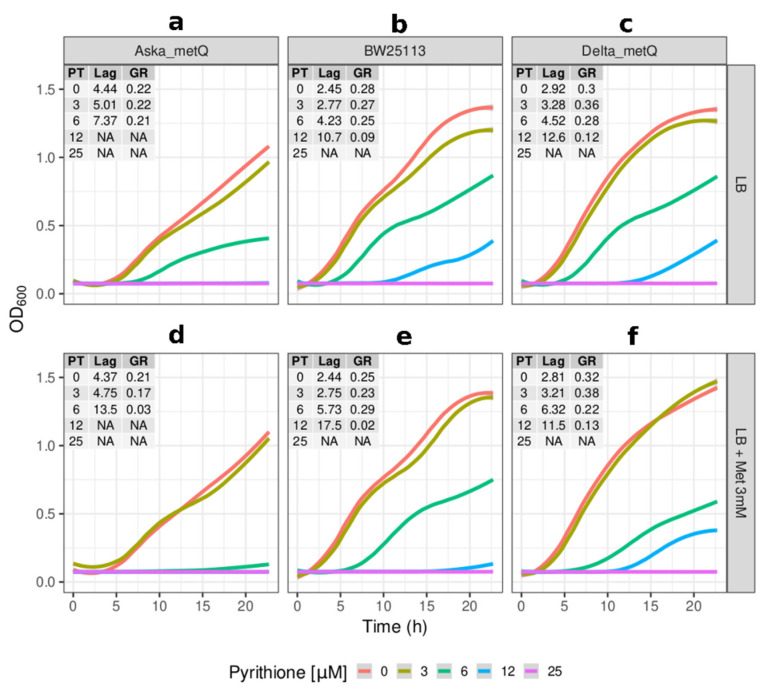
**Effects of MetQ in the sensitivity of *E. coli* to pyrithionine.** (**a**–**c**) Growth (OD_600_) at incremental concentrations of pyrithione in complex media (LB) of the Aska strain carrying an episomal copy of *metQ* (Aska_metQ) (**a**), the reference strain *E. coli* BW25113 (**b**); and the Keio strain of the *metQ* knockout (*Delta_metQ*) (**c**). (**d**–**f**) As above with the addition of methionine at 3 mM final concentration. Insets: PT: concentration of pyrithione in micromolar (µM), Lag: length of lag phase in hours, GR: growth rates fitted from optical densities of the cultures in exponential phases with the parametric non-linear method of least squares. The colour coded lines represent the fitted data for cultures under the given concentrations of pyrithione in micromolar (µM) units. NA (non-applicable): data below detection limit.

**Figure 5 molecules-26-05826-f005:**
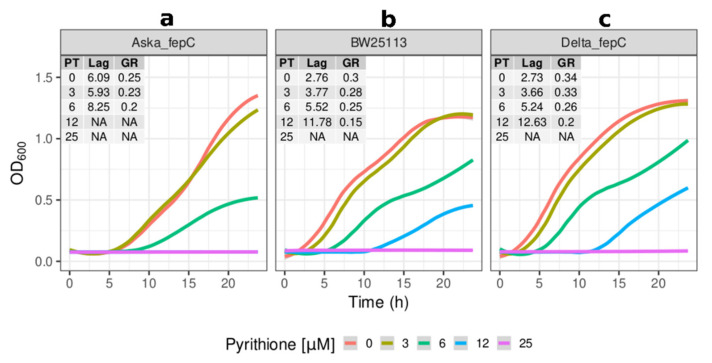
**Effects of *fepC* in the sensitivity of *E. coli* to pyrithione.** Growth (OD_600_) at incremental concentrations of pyrithione in complex media (LB) of the (**a**) Aska clone carrying episomal copies of *fepC* (Aska_fepC), (**b**) the reference strain BW25113, and (**c**) the *fepC* knockout strain (Delta_fepC). Insets: PT: concentration of pyrithione in micromolar (µM), Lag: length of lag phase in hours, GR: growth rate fitted from optical densities of the cultures in exponential phases. The colour coded lines represent the fitted data with the parametric non-linear method of least squares. NA (non-applicable): data below detection limit.

**Figure 6 molecules-26-05826-f006:**
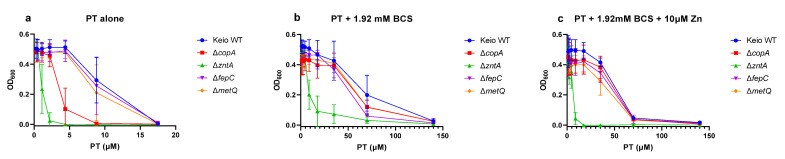
**Growth under incremental concentrations of pyrithione (PT).** Growth (OD_600_) of *E. coli* K-12 BW25113 (Keio WT) and the Keio knockout strains *ΔcopA*, *ΔzntA*, *ΔfepC*, and *ΔmetQ*, compared across the relevant concentrations of PT alone (**a**), or in combination with 1.92 mM BCS (**b**), or in combination with 1.92 mM BCS and 10 µM ZnCl_2_ (**c**).

**Figure 7 molecules-26-05826-f007:**
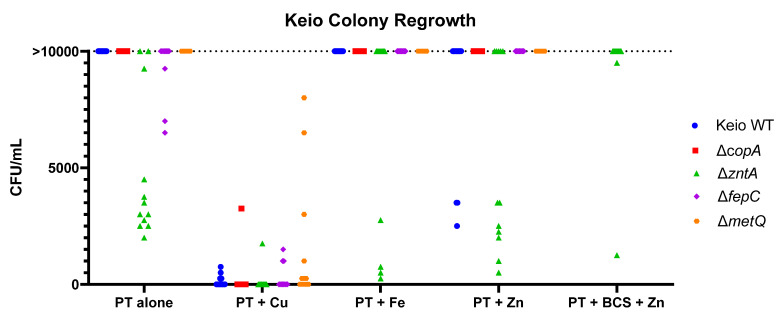
**Ability of PT to kill *E. coli* K-12 BW25113 (Keio WT) and the knockout strains, *ΔcopA*, *ΔzntA*, *ΔfepC*, and *ΔmetQ*.** Bacteria treated for 20 h with the minimum inhibitory concentration of pyrithione for that condition (Table 3) with or without 10 µM Cu, Fe, Zn, or 1.92 mM BCS, were plated on fresh LB agar and incubated at 37 °C for 24 h prior to colony enumeration. Each point on the graph represents an individual replicate. Number of viable cells (measured as CFU/mL) at 0 h for the untreated bacteria samples were as follows: Keio WT = 1.7 × 10^6^, *ΔcopA* = 1.3 × 10^6^, *ΔzntA* = 1.4 × 10^6^, *ΔfepC* = 0.7 × 10^6^, and *ΔmetQ* = 2.0 × 10^6^.

**Figure 8 molecules-26-05826-f008:**
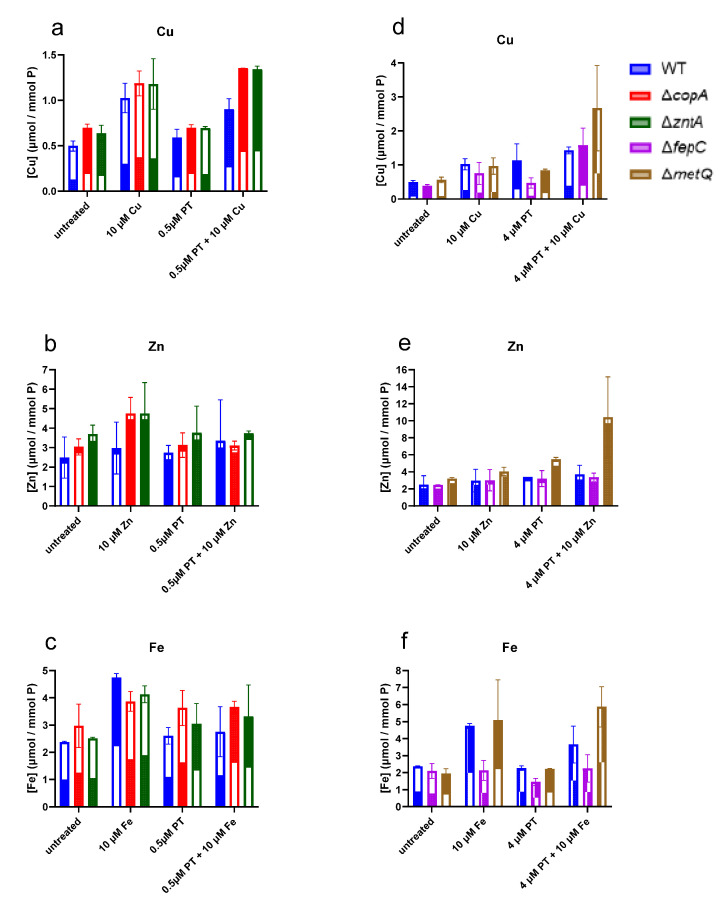
**Cell-associated Cu, Zn, and Fe in wild-type (WT) *E. coli* K-12 BW25113 and the Keio knockout strains.** (**a**) Cell-associated copper, (**b**) zinc, and (**c**) iron in the WT and knockout strains *ΔcopA* and *ΔzntA*. (**d**) Cell-associated copper, (**e**) zinc, and (**f**) iron in WT and knockout strains *ΔfepC* and *ΔmetQ*. Metal content was measured using ICP-MS as detailed in Materials and Methods and normalised to phosphorus for each sample. Data are shown as the average of two biological replicates taken with three technical replicates each, with the range indicated by the error bars.

**Table 1 molecules-26-05826-t001:** **Pyrithione inhibitory concentrations for the Keio collection *E. coli* reference strain, BW25113**. Cells growing in complex media (LB) were subjected to varying pyrithione concentrations in the presence of incremental concentrations of CuCl_2_. Inhibitory concentrations as IC_50_ values are shown as the mean and standard deviation of four replicates.

Media	CuCl_2_ (µM)	IC_50_ (µM)
Lysogeny Broth (LB)	0	6.05 ± 1.02
	5	5.22 ± 0.95
	10	0.78 ± 0.25
	20	0.81 ± 0.28

**Table 2 molecules-26-05826-t002:** **Growth rate ratios of the top and bottom ten hits from the subset of the Keio gene knockout collection exposed to pyrithione.** E. coli reference strain BW25113 (WT) divides the knockout (KO) strains presenting ratios of growth rates above and below the units. All strains were cultured in parallel in LB with or without 2 µM pyrithione. Full list available in Appendix A.

Strain (KO)	Growth Rate Ratio		Gene Annotation
	Mean	SD	
*metQ*	2.41	0.92	L-methionine/D-methionine ABC transporter membrane anchored binding protein
*fepC*	2.12	0.90	ferric enterobactin ABC transporter ATP binding subunit
*chbA*	1.75	0.70	*N,N’*-diacetylchitobiose-specific PTS enzyme IIA component
*araJ*	1.67	0.64	putative transport protein AraJ
*phoR*	1.66	0.46	sensor histidine kinase
*lldP*	1.59	0.41	lactate/glycolate:H^+^ symporter LldP
*potB*	1.59	0.46	spermidine preferential ABC transporter membrane subunit PotB
*glvB*	1.54	0.45	putative PTS enzyme II component GlvB
*hofC*	1.54	0.38	inner membrane protein HofC
*mdtC*	1.52	0.31	multidrug efflux pump RND permease subunit MdtC
**WT**	**1.04**	**0.11**	***E. coli* BW25113 (Keio collection reference strain)**
*tatC*	0.70	0.11	twin arginine protein translocation system-TatC protein
*yjeP*	0.70	0.13	mini conductance mechanosensitive channel MscM
*fucP*	0.69	0.19	L-fucose:H^+^ symporter
*ybiT*	0.69	0.14	putative ATP-binding protein YbiT
*yheS*	0.60	0.20	putative ATP-binding protein YheS
*malF*	0.54	0.10	maltose ABC transporter membrane subunit MalF
*xylE*	0.52	0.16	D-xylose:H^+^ symporter
*atpA*	0.44	0.12	ATP synthase F_1_ complex subunit &alpha;
*copA*	0.44	0.12	Cu^+^ exporting P-type ATPase (copA;ybaR;atcU)
*zntA*	0.09	0.03	Zn^2+^/Cd^2+^/Pb^2+^ exporting P-type ATPase

**Table 3 molecules-26-05826-t003:** Minimum inhibitory concentrations (MICs) of pyrithione (PT) against *E. coli* BW25113 and knockout strains in the presence of metals ^a^.

Conditions	BW25113	*metQ*	*fepC*	*copA*	*zntA*
**PT alone**	17.5	17.5	17.5	8.8	2.2
**PT and 10 µM Cu**	17.5	17.5	17.5	2.2	8.8
**PT and 10 µM Fe**	17.5	17.5	17.5	8.8	2.2
**PT and 10 µM Zn**	17.5	8.8	17.5	8.8	1.1
**PT and 1.92 mM BCS**	>140	>140	>140	>140	140
**PT and 1.92 mM BCS and 10 µM Zn**	>140	>140	140	>140	17.5

^a^ See Section 5.

## Data Availability

Data for growth of full Keio subset are in Appendix A. Further raw data are available from corresponding authors.

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
