# Peer review of "Membrane Transporters Involved in the Antimicrobial Activities of Pyrithione in Escherichia coli"

_molecules, 2021, doi:10.3390/molecules26195826_

Round 1
Reviewer 1 Report
General comments
The manuscript should be proofread for language errors and inconsistency in some formats. For example, the use of “hours” and “h” interchangeably.
The results section needs to be restructured. It is lengthy, mainly because the authors are discussing some of the aspects within this section. The authors should go straight to what was found without explaining what it means. Therefore, the results section cannot have references. All such explanations should be moved to the discussion section. E.g., Line 64-66, 90-110, etc.
Also, the authors seem to repeat the methods (saying what was done) before presenting the results under each section. Again, kindly go straight to the reporting what was found.
The conclusion should not have references.
Specific comments
Line 17: Change “interrogation” to “analysis”
Line 27: Remove the “e.g.” and reference properly. Do the same in many other instances in the manuscript.
Line 34-38: “In that work,….. [34,35]”. Please rephrase. This is long and complex to understand. I suggest splitting into shorter sentences.
Line 41: Change “capacity” to “ability”
Line 227: Remove )
Line 249: …taking photographs…
Line 279: What was the rationale of incubating LB agar plates under rotatory conditions (200 rpm). This would be understood for the broth, but why the agar plates also?
Line 281: Delete “via”
Author Response
Referee ONE
General comments
The manuscript should be proofread for language errors and inconsistency in some formats. For example, the use of “hours” and “h” interchangeably.
ANSWER
The manuscript has gone through further extensive proofreading as recommended by the referee. Thank you.
The results section needs to be restructured. It is lengthy, mainly because the authors are discussing some of the aspects within this section. The authors should go straight to what was found without explaining what it means. Therefore, the results section cannot have references. All such explanations should be moved to the discussion section. E.g., Line 64-66, 90-110, etc.
ANSWER
Sections of the Results have been merged in the Discussion as recommended. The rest we leave to the Editor’s decision.
Also, the authors seem to repeat the methods (saying what was done) before presenting the results under each section. Again, kindly go straight to the reporting what was found.
ANSWER
This is how it is expected to be set out. We leave this for the Editor.
The conclusion should not have references.
ANSWER
Although references in the Conclusions are not recommended, we are citing one reference necessary for the quoted sentence included in that section of the paper. Stylistic matters are for the Editor.
Specific comments
Line 17: Change “interrogation” to “analysis”
ANSWER
We are confident that “interrogation” described the intended meaning.
Line 27: Remove the “e.g.” and reference properly. Do the same in many other instances in the manuscript.
ANSWER
Changes carried out as recommended by the referee.
Line 34-38: “In that work,….. [34,35]”. Please rephrase. This is long and complex to understand. I suggest splitting into shorter sentences.
ANSWER
Sentence modified as suggested. Thank you.
Line 41: Change “capacity” to “ability”
ANSWER
Sentence modified as suggested. Thank you
Line 227: Remove )
ANSWER
Removed
Line 249: …taking photographs…
ANSWER
This text was modified as follows:
“These plates were sandwiched with plastic covers (CR1384, Enzyscreen) and incubated for 24 h at 37ºC with 225 rpm shaking in the Growth Profiler. This instrument scans individual wells and registers photographic records of them, set to be taken every 15 min.
Line 279: What was the rationale of incubating LB agar plates under rotatory conditions (200 rpm). This would be understood for the broth, but why the agar plates also?
ANSWER
Although rotatory conditions are not necessary for LB agar plates others in the lab were using the incubators for cultures that did require rotatory conditions. Therefore, they were subjected to 200 rpm rotatory conditions. Although this was not necessary, it also does not change the results in any way.
Line 281: Delete “via”
ANSWER
Changed as suggested.
Reviewer 2 Report
After reading the manuscript entitled "Membrane transporters involved in the antimicrobial activities 2 of pyrithione in Escherichia coli", I consider it appropriate to be published in Molecules.
The subject matter is original and important. Introduction section presents background information, this part is concise and well‐written. Materials and Methods section requires minor corrections. Discussion is written in an interesting way. References are correlated well with the text.
The changes in the text must be performed to increase its quality.
Abstract
page 1, line 16 is: … difussion…, should be: … diffusion…
Introduction
page 1, line 27 is: … (e.g. [1-11])…, should be: … [1–11]…
page 1, line 28 is: … (e.g. [12-21])…, should be: … [12–21]…
page 1, line 29 is: … (e.g. [22-26])…, should be: … [22–26]…
page 2, line 49 is: … (e.g. [45-51])…, should be: … [45–51]…
Results
page 2, line 62 is: … concentations…, should be: … concentrations …
page 3, line 102 is: … [73]]…, should be: … [73] …
page 3, line 113 is: … represenetd…, should be: … represented …
page 4 is: … Overexpresing copA…, should be: … Overexpressing copA …
page 6 is: … confouding…, should be: … confounding…
page 6 is: … overexpresion of MetQ…, should be: … overexpression of MetQ…
page 7 is: … an MIC lower…, should be: … a MIC lower…
page 7 is: … effect on MIC…, should be: … effect on the MIC…
page 7 is: … with an MIC value…, should be: … with the MIC value…
page 8 is: Table 3: Minimum…, should be: Table 3. Minimum…
page 8 is: “MICs were measured…….chelator” - the text under the table 3 should be moved to Materials and Methods section.
page 8 is: Figure 6: Growth…, should be: Figure 6. Growth…
page 8 is: “OD600 readings…….rpm” - this sentence in the figure 6 description should be moved to Materials and Methods section.
page 9 is: Figure 7: Ability…, should be: Figure 7. Ability…
page 9 is: “Bacteria……. 2.0 x 106” - this sentence in the figure 7 description should be moved to Materials and Methods section.
page 9 is: … associated to cells…, should be: … associated with cells…
Discussion
page 10, line 130 is: … tranport…, should be: … transport…
page 11, line 143 is: … identifiying…, should be: … identifying…
page 11, line 146 is: … muliple…, should be: … multiple…
page 11, line 174 is: … cause release…, should be: … cause the release…
page 11, line 182 is: … are consistent…, should be: … is consistent…
page 11, line 191 is: … methione…, should be: … methionine…
page 12, line 208 is: … absence…, should be: … the absence…
page 12, line 213 is: … straighforward…, should be: … straightforward …
page 12, line 214 is: … a iron…, should be: … an iron…
Conclusion
page 12, line 223 is: … tranported…, should be: … transported…
Materials and Methods
Authors should divide this section into subsections and entitle them.
page 13, lines 243, 248 is: … 24 hours.…, should be: … 24 h….
Reference
Bacteria names should be in italics.
Author Response
Referee TWO
After reading the manuscript entitled "Membrane transporters involved in the antimicrobial activities 2 of pyrithione in Escherichia coli", I consider it appropriate to be published in Molecules.
The subject matter is original and important. Introduction section presents background information, this part is concise and well‐written. Materials and Methods section requires minor corrections. Discussion is written in an interesting way. References are correlated well with the text.
The changes in the text must be performed to increase its quality.
Abstract
page 1, line 16 is: … difussion…, should be: … diffusion…
Introduction
page 1, line 27 is: … (e.g. [1-11])…, should be: … [1–11]…
page 1, line 28 is: … (e.g. [12-21])…, should be: … [12–21]…
page 1, line 29 is: … (e.g. [22-26])…, should be: … [22–26]…
page 2, line 49 is: … (e.g. [45-51])…, should be: … [45–51]…
Results
page 2, line 62 is: … concentations…, should be: … concentrations …
page 3, line 102 is: … [73]]…, should be: … [73] …
page 3, line 113 is: … represenetd…, should be: … represented …
page 4 is: … Overexpresing copA…, should be: … Overexpressing copA …
page 6 is: … confouding…, should be: … confounding…
page 6 is: … overexpresion of MetQ…, should be: … overexpression of MetQ…
page 7 is: … an MIC lower…, should be: … a MIC lower…
page 7 is: … effect on MIC…, should be: … effect on the MIC…
ANSWER
All the above changes were carried out through proofreading and spellchecking.
page 7 is: … with an MIC value…, should be: … with the MIC value…
ANSWER
Corrected. Thank you.
page 8 is: Table 3: Minimum…, should be: Table 3. Minimum…
ANSWER
Corrected. Thank you.
page 8 is: “MICs were measured…….chelator” - the text under the table 3 should be moved to Materials and Methods section.
ANSWER
Changed as suggested.
page 8 is: Figure 6: Growth…, should be: Figure 6. Growth…
ANSWER
Corrected. Thank you.
page 8 is: “OD600 readings…….rpm” - this sentence in the figure 6 description should be moved to Materials and Methods section.
ANSWER
Changed as suggested.
page 9 is: Figure 7: Ability…, should be: Figure 7. Ability…
ANSWER
Corrected.
page 9 is: “Bacteria……. 2.0 x 106” - this sentence in the figure 7 description should be moved to Materials and Methods section.
ANSWER
The data described is in context and necessary for the results presented in this figure.
page 9 is: … associated to cells…, should be: … associated with cells…
ANSWER
Changed as suggested.
Discussion
page 10, line 130 is: … tranport…, should be: … transport…
page 11, line 143 is: … identifiying…, should be: … identifying…
page 11, line 146 is: … muliple…, should be: … multiple…
page 11, line 174 is: … cause release…, should be: … cause the release…
page 11, line 182 is: … are consistent…, should be: … is consistent…
page 11, line 191 is: … methione…, should be: … methionine…
page 12, line 208 is: … absence…, should be: … the absence…
page 12, line 213 is: … straighforward…, should be: … straightforward …
page 12, line 214 is: … a iron…, should be: … an iron…
Conclusion
page 12, line 223 is: … tranported…, should be: … transported…
ANSWER
All the above changes in Discussion and Conclusions were carried out through proofreading and spellchecking.
Materials and Methods
Authors should divide this section into subsections and entitle them.
ANSWER
Yes, we agree. This section is now modified accordingly.
page 13, lines 243, 248 is: … 24 hours.…, should be: … 24 h….
ANSWER
Changed as suggested.
Reference
Bacteria names should be in italics.
ANSWER
Corrected. Thank you.
Reviewer 3 Report
Pyrithione is a well-known antibacterial and antifungal agent with multiple applications used for over 60 years. However, the precise mechanism of its action remains elusive, particularly the cellular uptake. The present study broadens the understanding of the mechanisms of action of pyrithione.
In their manuscript, Salcedo-Sora et al. have experimentally identified two transporters, namely FepC and MetQ, that can be involved in the uptake of pyrithione and its complexes with metals. Moreover, the results presented in this manuscript also revealed the importance of two metal efflux pumps, CopA and ZntA, in counteracting the hyperaccumulation of copper and zinc inside the bacterial cells due to the pyrithione action.
The manuscript is well-written, the chosen methodology does not raise questions, and all conclusions are firmly supported by the results. I recommend publishing this manuscript in its present form.
Author Response
Referee THREE
Pyrithione is a well-known antibacterial and antifungal agent with multiple applications used for over 60 years. However, the precise mechanism of its action remains elusive, particularly the cellular uptake. The present study broadens the understanding of the mechanisms of action of pyrithione.
In their manuscript, Salcedo-Sora et al. have experimentally identified two transporters, namely FepC and MetQ, that can be involved in the uptake of pyrithione and its complexes with metals. Moreover, the results presented in this manuscript also revealed the importance of two metal efflux pumps, CopA and ZntA, in counteracting the hyperaccumulation of copper and zinc inside the bacterial cells due to the pyrithione action.
The manuscript is well-written, the chosen methodology does not raise questions, and all conclusions are firmly supported by the results. I recommend publishing this manuscript in its present form.
ANSWER
Thank you.
Round 2
Reviewer 1 Report
Responses noted